# Study of a Novel Electrochromic Device with Crystalline WO_3_ and Gel Electrolyte

**DOI:** 10.3390/polym14071430

**Published:** 2022-03-31

**Authors:** Wanyu Chen, Guixia Zhang, Lili Wu, Siyuan Liu, Meng Cao, Ying Yang, Yong Peng

**Affiliations:** 1School of Materials Science and Engineering, Wuhan University of Technology, Wuhan 430070, China; chenwanyu@whut.edu.cn (W.C.); zhang94670@whut.edu.cn (G.Z.); polym_wl@whut.edu.cn (L.W.); liusiyuan_ysl@whut.edu.cn (S.L.); caomeng19970705@163.com (M.C.); yingyang2006@163.com (Y.Y.); 2State Key Laboratory of Advanced Technology for Materials Synthesis and Processing, Wuhan University of Technology, Wuhan 430070, China

**Keywords:** electrochromic device, crystalline tungsten oxide, ionically cross-linked gel, conductivity

## Abstract

Most ECDs are coated with an electrochromic material on the transparent conductive oxide (TCO) substrate. A novel electrochromic device (ECD), having a variable optical performance, was prepared by using tungsten foil as a substrate in this study. It was found that the WO_3_ discoloration layer, having a monoclinic phase crystalline structure made of 600 °C calcined, had optimum charge transmission performance with PADA gel polymer electrolyte. Ionic conductivity of PADA gel polymer electrolyte was 2.3 × 10^−3^ S cm^−1^ at −20 °C, and it was possible to help Li^+^ to implement embedding and extraction from WO_3_ even in low-temperature conditions. The colored time (t_c_) and the bleached time (t_b_) of the electrochromic device were 15 s and 26 s, and it showed yellowish-brown in the colored state and navy blue in the bleached state. The ECD (WO_3_-600) exhibited good cycle stability reach at least 150 times.

## 1. Introduction

Electrochromism refers to a reversible change in the electronic structure and optical properties (reflectance, transmittance or absorption) of electrochromic materials under an applied voltage [1,2,3,4]. Electrochromic phenomena have a wide range of applications in the fields of smart windows, displays, anti-glare rearview mirrors, smart thermal control and camouflage, etc., [5,6,7,8]. WO_3_, an electrochromic layer material, has been widely used in inorganic electrochromic and other related technical fields because of its advantages, such as high-color contrast, low manufacturing cost, non-toxicity, and stability under acidic and oxidizing conditions [9,10,11,12,13,14,15,16,17].

Normally, WO_3_ layers are prepared on top of transparent conductive oxide (TCO) via hydrothermal method, sol–gel progress, sputtering, etc. [18,19,20,21], and ECD mostly consists of TCO/cathode electrode/electrolyte/anode electrode. However, TCO is not cheap, and the preparation process of WO_3_ is relatively complex. For example, Pooyodying, et al., focused on the coating of WO_3_ on PET/ITO substrates [22]. In addition, there are different phases of WO_3,_ such as orthorhombic, monoclinic (m-WO_3_), triclinic, tetragonal (t-WO_3_), hexagonal (h-WO_3_), etc., [23,24,25,26,27]. The structure of WO_3_ prepared by these synthesis methods is mostly orthorhombic and monoclinic phase [23,28,29]. In this paper, a novel method was the preparation of the WO_3_ layer on tungsten foil via direct oxidization at high temperatures. The WO_3_ had a porous structure and three crystal phases after calcining, and its new performance application was given. The porous crystalline WO_3_ film had high porosity, specific surface area, better stability, and higher strength and corrosion resistance [30], which affected charge carriers transporting during working. Three different crystals of hexagonal, tetragonal and monoclinic phase of WO_3_ on EC properties were investigated.

As the ion transport layer, the electrolyte is located in the middle layer of the electrochromic device, and is the conductive medium between the electrodes. Therefore, it is very important that the electrolyte has good performance in terms of electrical conductivity and electrochemical stability. In addition, the electrolyte used needs to have a high transmittance to effectively avoid influencing the color modulation of the electrochromic material. Gel polymer electrolyte combines the advantages of liquid electrolyte and solid electrolyte with good flexibility, high ionic conductivity, and a wide electrochemical stability window, which has made it widely studied, such as when PVDF-co-HFP/[Li][TFSI] gel electrolytes were designed for flexible WO_3_-based ECDs [31].

In this research, the prepared PADA gel electrolyte had a high-transmittance, high-ion conductivity and a wide electrochemical stability window. The compatibility of the electrochromic layer with the ion transport layer also affected the performance of the electrochromic device. An electrochromic device composed of copper-conductive adhesive/tungsten oxide foil/gel electrolyte/nickel foil/transparent PET film was assembled. The three crystal phases of the tungsten oxide and the PADA gel electrolyte were assembled into an electrochromic device (referred to as ECD (WO_3_-400), ECD (WO_3_-500), ECD (WO_3_-600)) by multi-sandwiching, and its optical performance and cycle stability was studied.

## 2. Experimental Section

### 2.1. Materials

N,N-dimethylformamide (DMF, AR) was purchased from Sinopharm Chemical Reagent Co., Ltd. (Shanghai, China). Lithium perchlorate (LiClO_4_, anhydrous, 99.9%), propylene carbonate (PC, anhydrous, 99.7%), acrylic acid (AA, purity > 99%), acrylamide (AAm, 99%) and 2-diethylaminoethyl methacrylate (DEAM, 99%) were purchased from Aladdin (Shanghai, China). Azobisisobutyronitrile (AIBN, AR) was provided by Sigma–Aldrich. (Shanghai, China). Tungsten (99.99%) was purchased from Xinyi Metal Co., Ltd. (Xingtai, China).

### 2.2. Synthesis of PADA Gel Polymer

The specific synthesis process was as follows [32]: firstly, 0.5032 g LiClO_4_ was added into a mixed solution of 2.5 mL of propylene carbonate (PC) and 2.5 mL of N,N-dimethylformamide (DMF), and stirred until the solution became transparent and clear. Secondly, 0.5 mL of diethylaminoethyl methacrylate (DEAM) and 0.5 mL of acrylic acid (AA) were added as ion associations into the mixed solution. Thirdly, 250 mg of comonomer acrylamide (AAm) was added. Fourthly, 1.25 mg of AIBN was added to induce initiation. Fifthly, by stirring at room temperature for 2 h, an homogeneous solution was obtained. Finally, the oxygen-free purity mixture was put into a clean and sealed glass container and heated at 80 °C for 8 h to form PADA gel electrolyte. Table 1 shows the proportion composition of different PADA gels.

### 2.3. Assembly of PECD

A clean tungsten foil was put in a crucible and was then transferred to a KSL-1100X box furnace for calcining. The oxidizing temperatures were set to a fixed value (such as 400 °C or 500 °C or 600 °C) for 3 h. Then, oxidized tungsten was prepared for device assembling. As shown in Figure 1, the electrochromic device was assembled to be a sandwich structure of copper-conductive adhesive/tungsten oxide foil/gel electrolyte/nickel foil/transparent PET film.

### 2.4. Characterization

The phase state tungsten of the tungsten oxide was analyzed by X-ray diffraction (XRD, D8 Advance X, Bruker, AXS, Karlsruhe, Germany), using Cu-Kα as the target, with a scanning range of 20–90° and a scanning speed of 10 mv/s.

The microstructure, morphology and atomic composition of the WO_3_ were characterized by using a field emission scanning electron microscope with an X-Max N80 spectrometer (FE-SEM, JSM-7500F, JEOL, Tokyo, Japan).

Optical reflectivity of the electrochromic device was measured by using an ultraviolet, visible, near-infrared spectrophotometer (Lambda 750S, PerkinElmer, Waltham, MA, USA) with a wavelength range of 400–700 nm.

An electrochemical workstation (SP300, Bio-Logic, Seyssinet-Pariset, France) was used to measure the electrochemical impedance spectroscopy (EIS) behavior of the PADA gel electrolyte at a frequency of 0.1 Hz–1 MHz and a temperature of −20–80 °C. The structure of the test device was Pt/PADA gel electrolyte/Pt. The equivalent circuit diagram of the gel electrolyte was obtained by fitting Zview software.

## 3. Results and Discussion

### 3.1. Optical Properties of Electrochromic Devices

Figure 2a,b show the colored state and the bleached state of ECD (WO_3_-600), respectively. ECD (WO_3_-600) shows yellowish-brown in the colored state and navy blue in the bleached state. The optical modulation amplitude is defined as the reflectivity difference between the colored state and the bleached state. Figure 2c–e show that the reflectivity of the colored state ECD (WO_3_-600) and the bleached state of ECD (WO_3_-600) under a voltage of ±3 V at 700 nm are 16.59% and 12.49%, respectively; the optical modulation amplitude is 4.1%. Compared with ECD (WO_3_-400) and ECD (WO_3_-500), ECD (WO_3_-600) has a larger optical modulation amplitude, which indicates that ECD (WO_3_-600) is more suitable as an electrochromic device.

The response time of an electrochromic device is defined as the time required for the electrochromic device to complete 90% of the maximum optical modulation amplitude. Figure 2f shows the coloring and bleaching time of different electrochromic devices. Compared to ECD (WO_3_-400) and ECD (WO_3_-500), the coloring time of ECD (WO_3_-600), whose color switch change rate is t_c_/t_b_ = 15/26, is shorter. The electrochromic device has a large optical modulation amplitude and color change speed: these are associated with the transparency and ion conductivity of the gel electrolyte.

Based on the transmittance test of the PADA gel electrolyte, synthesized with different DEAM and AA volume ratios in the wavelength range of 400–800 nm, Figure 3 shows that the transparency of the PADA-b gel electrolyte is higher than that of PADA-a and PADA-c. This may be the reason why the volume ratio of the AA and DEAM is 1:1, the compatibility between the polymer and the solvent is optimal—forming a compatible gel system which exhibits high transparency [32]—and its optical transmittance exceeds 98% in the range of 610–800 nm, and reaches 99.88% at 800 nm. However, the transmittance of some organic polymer gel electrolytes such as PEO and PI is within 80–90% [33,34,35,36,37,38]. The high transmittance of the gel electrolyte is better used in electrochromic devices; the difference in transmittance or reflectivity between the colored state and the bleached state of the electrochromic device can be clearly observed.

We conducted electrical impedance testing of the synthesized PADA-a, PADA-b, PADA-c and liquid electrolyte to analyze their ionic conductivity. Ionic conductivity plays an important role in determining the efficiency of ion transport in the electrolyte, which greatly affects the performance of electrochromic devices. As the ion-transport efficiency of the electrolyte has a great influence on the response time of the coloring and fading process of the electrochromic device, it is also one of the major indicators of the performance of the ECD.

Figure 4a shows the AC impedance spectra of different PADA gel electrolytes and liquid electrolytes at 20 °C. Figure 4b shows that arc was hardly observed in the high frequency part of the AC impedance spectrogram, so a constant phase angle element (CPE) was used instead of the capacitor in the equivalent circuit diagram [39,40]. Figure 4c is the equivalent circuit diagram of the electrolyte that was fitted by Zview software; *R_s_* is the bulk resistance of the electrolyte; *R_f_* is the contact resistance between the gel electrolyte and the platinum electrode. In addition, *R_s_* was fitted by Zview software [41]. The conductivity value was calculated according to the following Equation (1):(1)$$\sigma =\frac{1}{R_{S}}\frac{l}{S}$$
where *l* is the distance between the two platinum electrodes and *S* is the contact area between the platinum electrode and the electrolyte. In this paper, *l*/*S* = 2 cm^−1^.

Figure 4d indicates the relationship between the temperature and the conductivity of a series of PADA gel electrolytes. It shows that the conductivity of gel electrolytes with different ratios increases with the increase of temperature. When the DEAM: AA is 1, the conductivity value of PADA-b gel is higher. The increase in ionic conductivity of the gel electrolyte is because Li^+^ in the gel moves faster as the temperature increases. Figure 4d also shows that the ionic conductivities of the PADA gel electrolyte and of the liquid electrolyte are in the same order of magnitude when compared, indicating that the conductivity of gel electrolyte is equivalent to that of liquid electrolyte. The ion conductivity of the PADA-b gel electrolyte is 6.79 × 10^−3^ S cm^−1^ at 20 °C, which is significantly superior to the PVDF-HFP and PEGDA chemical cross-linked gel polymer electrolyte [42,43] (10^−5^–10^−3^ S cm^−1^ at room temperature). In addition, the ionic conductivity of PADA-b gel electrolyte is 2.3 × 10^−3^ S cm^−1^ at −20 °C, indicating that PADA-b gel electrolyte still has a high conductivity at lower temperatures, which extends the temperature range of ECD working normally. This shows that PADA-b gel electrolyte can quickly transport Li^+^ when used as an ion-transport layer in electrochromic devices.

We established a relationship between the ionic conductivity and the temperature of the gel electrolyte according to the Arrhenius Equation (2) [41]:(2)$$\sigma =Aexp\left(\frac{-E_{a}}{KT}\right)$$
where *σ* is the ionic conductivity, *A* is the frequency factor, *E_a_* is the apparent activation energy, *K* is the rate constant and *T* is the thermodynamic temperature. Figure 4d shows that the ionic conductivity of a series of PADA gel electrolytes increases with the increase of temperature, indicating that the conduction behavior of the PADA gel electrolyte follows the Arrhenius equation. It is a fact that polymer chains are easier to move at high temperatures, which makes Li^+^ transport easier. This shows that the migration movement of Li^+^ wrapped in polymer electrolyte is controlled by temperature. Therefore, the conductive behavior of PADA gel electrolyte is similar to liquid electrolyte.

### 3.2. Cyclic Stability of Electrochromic Devices

Figure 5a shows the repeated coloring and bleaching process of ECD (WO_3_-600) during 150 cycles. When the ECD (WO_3_-600) shown in Figure 5b was cycled for the first 11 times, itscolored reflectivity increased, starting from the 12th time initially; the R_c_ of the electrochromic device remained at about 18.67%; and the R_b_ remained unchanged at about 14.62%. After 150 bleach/coloring cycles, the R_c_ and R_b_ of the electrochromic device were 18.77% and 13.90%, respectively, which indicates that ECD(WO_3_-600) has good cycle stability at least for 150 cycles, and that this device retained 95% of its initial performance. This is because the stability performance of electrochromic devices is mainly affected by the stability of the gel electrolyte and the discoloration stability of WO_3_.

As shown in Figure 6a, when the applied voltage is lower than 3 V, the current through the PADA gel electrolyte is basically stable, which indicates that the gel electrolyte can maintain electrochemical stability at a voltage of 3 V. When the applied voltage is greater than 3 V, the current of the gel electrolyte increases with the increase of the voltage. When the applied voltage reaches 3.4 V, the passing current has increased dramatically, which means that the gel electrolyte begins to decompose at that point, so the electrochemical window of the gel electrolyte is 3.4 V. Figure 6b is the chronocurrent curve of the electrochromic device after sealing the gel electrolyte in the electrochromic device under the voltage condition of ±3 V. This indicates that the current through the electrochromic device is stable in the coloring/fading state, which also confirms PADA-b gel can maintain electrochemical stability under 3 V voltage, and that the corresponding electrochromic device has better stability performance.

As mentioned previously, crystalline WO_3_ was prepared and applied to electrochromic devices (Figure 1). Tungsten oxide foil was prepared by changing the calcination temperature and controlling other reaction conditions under the same conditions, to explore tungsten oxides with different crystallinity on the electrochromic device. The surface structure of tungsten foil and tungsten oxide foil prepared at different calcination temperatures was analyzed by XRD test (Figure 7a). Figure 7a shows that pure tungsten foil has a long sharp peak at the diffraction angle of 58.1°, and that the other peaks are hardly discovered. WO_3_-400 also has a small peak at the diffraction angle of 75.4°, indicating that the surface of the tungsten foil has been oxidized. These two peaks match the characteristic peaks in the hexagonal phase of WO_3_ (JCPDS NO.33-1387). Compared with WO_3_-400, WO_3_-500 has a short and broad peak at a diffraction angle of 23.6°, which is consistent with the characteristic peak in WO_3_ tetragonal phase (JCPDS NO.05-0388). It is difficult to observe at a diffraction angle of 75.4° in the XRD diagram of WO_3_-600, that the diffraction angle is 23.6° long spikes, and there is a small peak in the diffraction angle of 48.1°, indicating that the sample has a monoclinic phase (JCPDS NO.43-1035) [44,45,46].

Figure 7b and Table 2 show SEM and EDS tests for tungsten foil and tungsten oxide foil, respectively. The results of EDS analysis show that the W/O atomic ratio is close to 1:3, indicating that high-temperature calcined tungsten foil contains WO_3_ and basically conforms to the stoichiometric ratio. Figure 7b shows that the surface of the tungsten foil has a micropore structure, and that the surface of the tungsten oxide foil has many microcracks, which provide a place for Li^+^ to be embedded/extracted. As shown in Figure 8a,d,g, the morphology of tungsten oxide foil prepared at different calcination temperatures is analogous. According to the atomic force microscope (AFM) inspections of W and WO_3_ surface morphologies in Figure 9, the micropore structure of tungsten oxide foil photographed by SEM is further confirmed. The possibility of different performance of electrochromic devices may reduce due to morphological differences. Therefore, the crystallinity of tungsten oxide may have an effect on the stability performance of the electrochromic device.

Figure 10 indicates that ECD (WO_3_-600) has good stability performance after 50 cycles compared to ECD (WO_3_-400) and ECD (WO_3_-500). The symmetry of the crystal structure of WO_3_ increases continuously from monoclinic to tetragonal to hexagonal crystal phase; the corresponding crystal structure becomes more and more stable [47,48]. In this paper, the structure of WO_3_-400 and WO_3_-500 are more stable compared to WO_3_-600, which is not conducive to discoloration; the rate of insertion/extraction of Li^+^ in the WO_3_ layer becomes slower. Therefore, WO_3_-600 has optimum charge transmission performance with the PADA gel polymer electrolyte. It is more suitable as an electrochromic layer, which is consistent with Figure 5b. Figure 5b shows that under the condition of a fixed wavelength, the maximum reflectance of ECD (WO_3_-600) coloring continuously increases with the number of cycles. This is because the internal microcracks of WO_3_-600 are wider at the beginning (Figure 8g), Li^+^ insertion/extraction is easier, and WO_3_-600 discoloration speed is fast. As the number of cycles increases to 12, the internal-microcracks-width of the tungsten oxide film is substantially stable, and the coloring/bleaching speed of the tungsten oxide film also becomes stable. The cycle stability of ECD (WO_3_-600) can reach at least 150 times, which indicates that the electrochromic device has better stability.

## 4. Conclusions

In this study, we developed a novel EC device, with an architecture of copper conductive adhesive/tungsten oxide foil/gel electrolyte/nickel foil/transparent PET film, which has no need to use an expensive TCO layer. The influences on phase and micro-structure of high-temperature calcined WO_3_ from tungsten foils on the charge-carrier transporting properties in the EC device were investigated. The PADA gel electrolyte had a transmittance of 99.12% at 700 nm, electrical conductivity of 2.3 × 10^−3^ S cm^−1^ at −20 °C, 6.79 × 10^−3^ S cm^−1^ at 20 °C, and an electrochemical stability window of 3.4 V. A monoclinic phase crystalline structure was found to be more suitable to this PADA electrolyte. The EC device with a yellowish-brown (color)/navy blue (bleached) state demonstrated a coloring time of 15 s and a bleaching time of 26 s. In addition, after 150 bleach/coloring cycles, this device retained 95% of its initial performance.

## Figures and Tables

**Figure 1 polymers-14-01430-f001:**
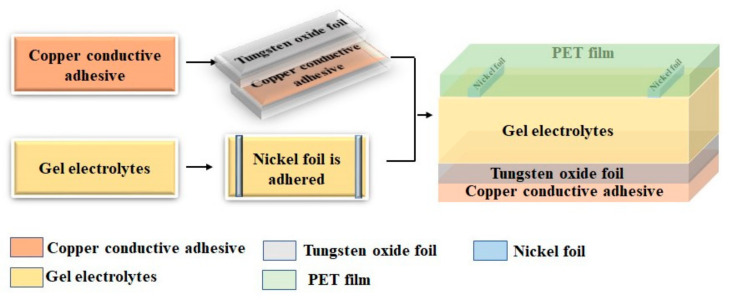
Assembly of PECD.

**Figure 2 polymers-14-01430-f002:**
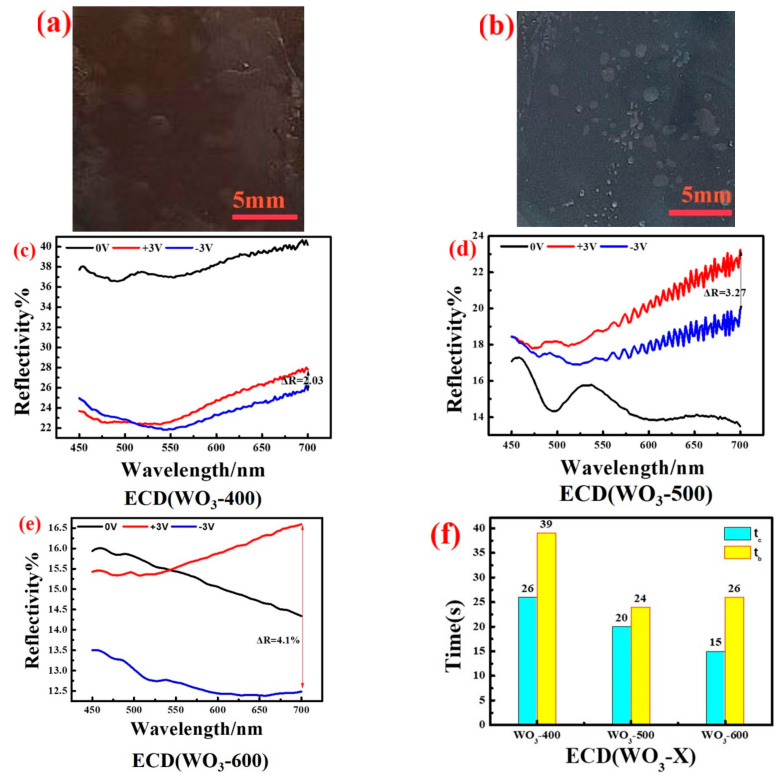
(**a**) Photo of the colored state of the electrochromic device; (**b**) Photo of the bleached state of the electrochromic device; (**c**) ECD reflectance spectrum of WO_3_-400 under different voltage conditions; (**d**) ECD reflectance spectrum of WO_3_-500 under different voltage conditions; (**e**) ECD reflectance spectrum of WO_3_-600 under different voltage conditions; (**f**) Coloring and bleaching time of ECD (WO_3_-X).

**Figure 3 polymers-14-01430-f003:**
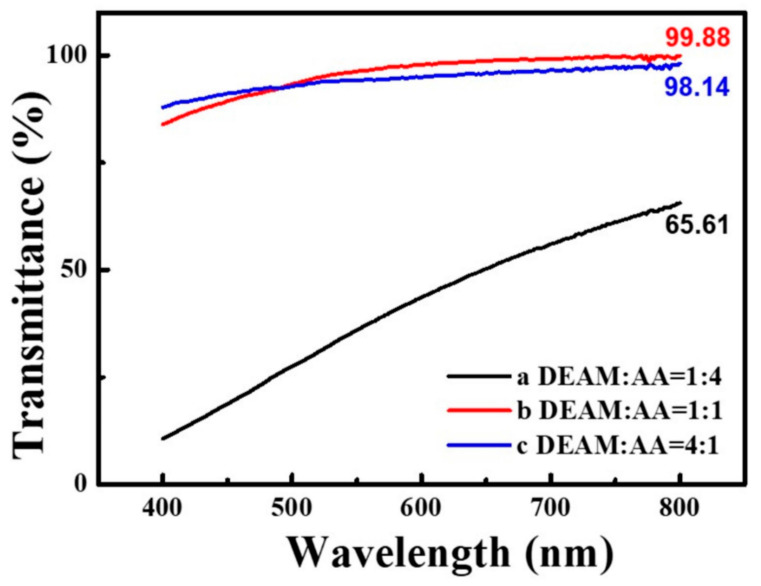
Transmittance spectra of PADA gel electrolytes with different volume ratios of DEAM and AA at wavelengths of 400–800 nm.

**Figure 4 polymers-14-01430-f004:**
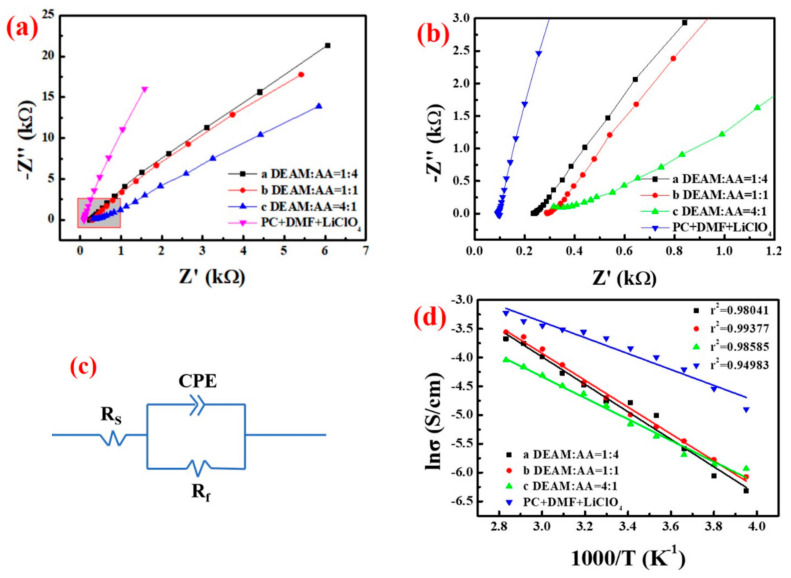
(**a**) AC impedance of PADA gel electrolytes with different volume ratios of DEAM and AA at 20 °C Spectrogram; (**b**) Enlarged view of the rectangular area in (**a**); (**c**) Equivalent circuit diagram of electrolyte fitted by Zview software; (**d**) The ionic conductivity of PADA gel electrolyte with different volume ratios of DEAM and AA changes with temperature.

**Figure 5 polymers-14-01430-f005:**
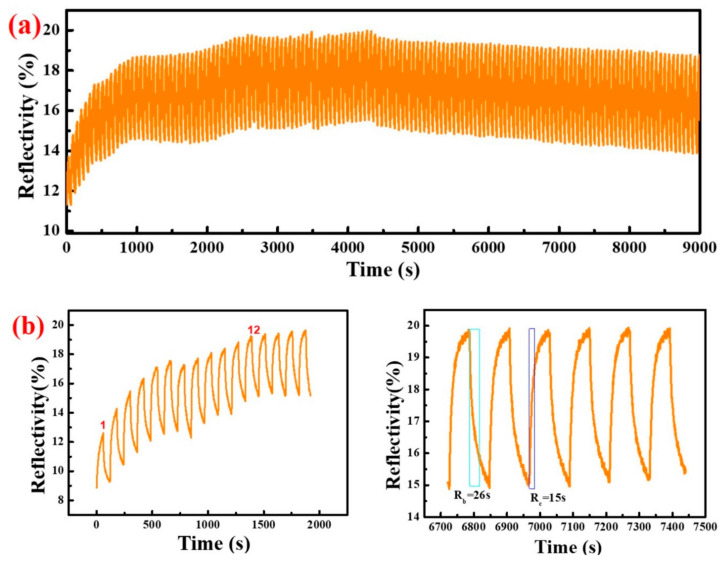
(**a**) The reflectance of the electrochromic device with WO_3_ at 700 nm for 150 cycles, applied voltage: ±3.0 V, time: 60 s; (**b**) Shows that the partial enlargement of (**a**): the left picture is 1–16th cycles, and the right picture is 56–62th cycles.

**Figure 6 polymers-14-01430-f006:**
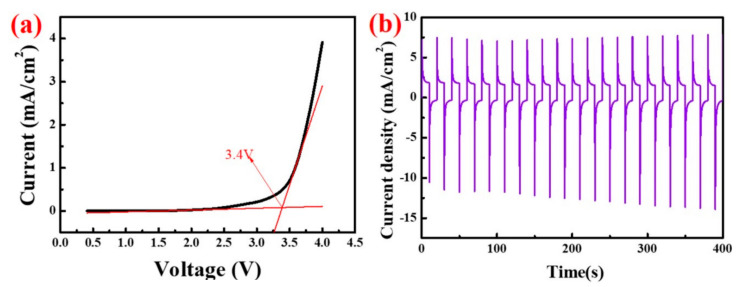
(**a**) Linear sweep volt-ampere curve of PADA-b gel electrolyte; (**b**) Chronoamperometry curve of the ECD (WO_3_-600) at a voltage of ±3.0 V.

**Figure 7 polymers-14-01430-f007:**
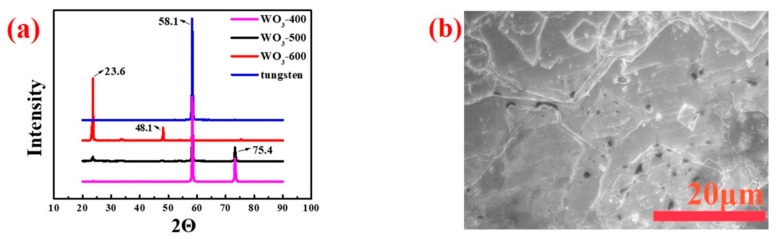
(**a**) XRD images of tungsten foil and tungsten oxide foil (WO_3_-X) calcined at different temperatures; (**b**) SEM images of tungsten foil.

**Figure 8 polymers-14-01430-f008:**
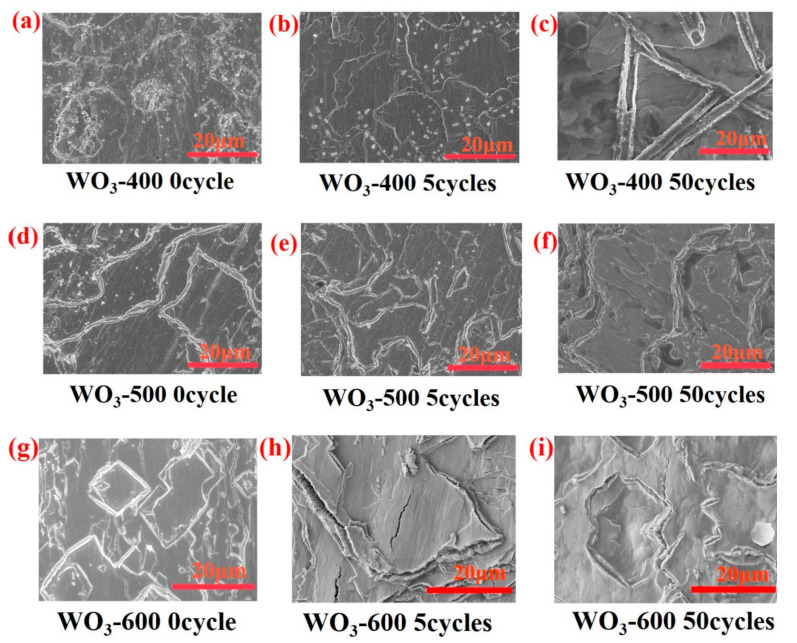
SEM images of WO_3_-X (X = 400, 500, 600). (**a**) ECD(WO_3_-400) after 0 cycle; (**b**) ECD(WO_3_-400) after 5 cycles; (**c**) ECD(WO_3_-400) after 50 cycles; (**d**) ECD(WO_3_-500) after 0 cycle; (**e**) ECD(WO_3_-500) after 5 cycles; (**f**) ECD(WO_3_-500) after 50 cycles; (**g**) ECD(WO_3_-600) after 0 cycle; (**h**) ECD(WO_3_-600) after 5 cycles; (**i**) ECD(WO_3_-600) after 50 cycles.

**Figure 9 polymers-14-01430-f009:**
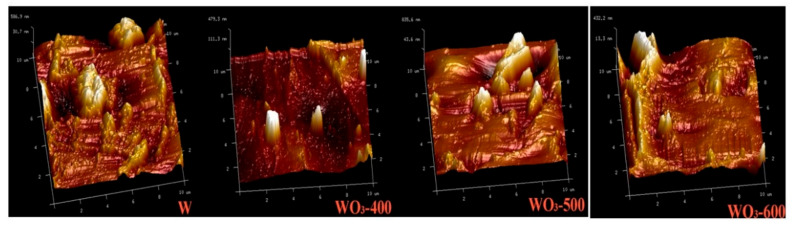
AFM images of W and WO_3_-X (X = 400, 500, 600).

**Figure 10 polymers-14-01430-f010:**
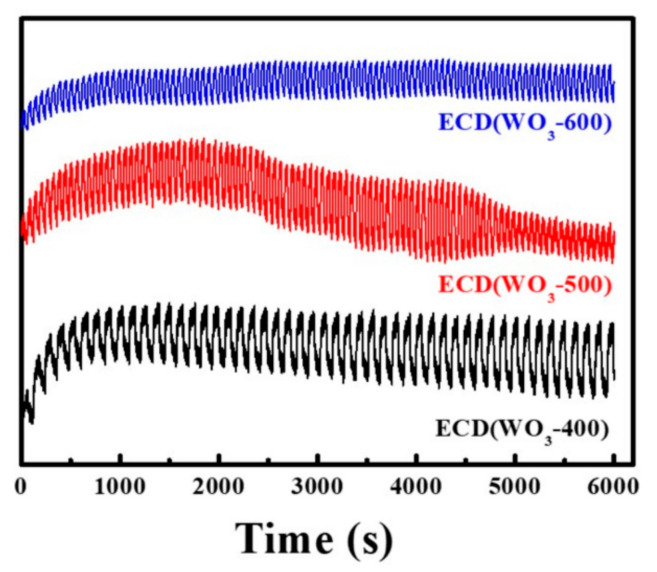
Circulating map of tungsten oxide foil prepared at different calcination temperatures for electrochromic devices, applied voltage: ±3.0 V; WO_3_-400 cycle time is 120 s; WO_3_-500 and WO_3_-600 cycle time are 60 s.

**Table 1 polymers-14-01430-t001:** Proportion composition of PADA gel electrolyte.

	PADA-a	PADA-b	PADA-c
PC (mL)	2.5	2.5	2.5
DMF (mL)	2.5	2.5	2.5
LiClO_4_ (g)	0.5032	0.5032	0.5032
DEAM (mL)	0.2	0.5	0.8
AA (mL)	0.8	0.5	0.2
AAm (mg)	250	250	250
AIBN (mg)	1.25	1.25	1.25

**Table 2 polymers-14-01430-t002:** Element content of W and O in EDS spectrum.

	Element	Tungsten	WO_3_-400	WO_3_-500	WO_3_-600
Weight%	O	0	16.69	21.21	31.32
	W	100	83.31	78.79	68.68
Atomic%	O	0	69.71	75.57	83.98
	W	100	30.29	24.43	16.02

## Data Availability

The data presented in this study are available on request from the corresponding author.

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
