# Peer review of "Study of a Novel Electrochromic Device with Crystalline WO3 and Gel Electrolyte"

_polymers, 2022, doi:10.3390/polym14071430_

Round 1
Reviewer 1 Report
The article by Wanyu Chen deals with a novel electrochromic device (ECD) prepared using tungsten foil as a substrate. The article is well written and suitable for the journal. The topic is fairl novel, though previous works dealing with similar devices based on WO3 have been previously reported. THerfore, the novelty of the current work should be more clearly highlighted, specially in the introduction. Besides, some other points should be fixed prior to possible publication in the journal:
1-Figure 1 is not very apealling for the reader. A more real and ellaborated scheme should be provided.
2- The scale in Figure 3 does not provide information. Please modify
3- Figure 7 is difficult to be interpreted. Should be magnified and axis expanded in order to be able to provide information to the reader.
4-Only one SEM image is not representative. More images should be provided in order to be able to provide a more accurate information. In addition, other microscopic techniques such as TEM or AFM images should be provided to show higher magnification images, hence higher resolution.
Author Response
We sincerely thank you all for spending precious time and efforts in examining this manuscript and greatly appreciate your insightful comments to make the paper better. The manuscript has been carefully revised, improved and verified to address the questions raised by the reviewers, and revisions have been marked with yellows for clarification. Please see the attachment.

Reviewer 2 Report
This works dealing with the Tungsten oxide-based electrochromic materials coupled with gel electrolyte can be interesting for some readers, but it definitely must be improved before further consideration.
1) Novelty. Authors must clearly show their innovations. Indeed, WO3 is old and well-known EC material. There are hundreds of publications, patents and working devices. Show clearly your improvements and state in the Abstract.
2) Introduction/ references: if you start your work on WO3 EC you cannot avoid the papers that are is fundamental basic for WO3 electrochromic devices. I am absolutely surprised, that not a single work from the shortlist below was not reviewed during your manuscript preparation. Please carefully study the following works for your introduction.
Granqvist, C.G.
Electrochromic tungsten oxide films: Review of progress 1993-1998
DOI: 10.1016/S0927-0248(99)00088-4
Granqvist, C.G.
Electrochromics for smart windows: Oxide-based thin films and devices
DOI: 10.1016/j.tsf.2014.02.002
Granqvist, C.G.
Oxide electrochromic: An introduction to devices and materials
DOI: 10.1016/j.solmat.2011.08.021
Granqvist, C.G., Green, S., Niklasson, G.A., Mlyuka, N.R., von Kræmer, S., Georén, P.
Advances in chromogenic materials and devices
DOI: 10.1016/j.tsf.2009.08.058
Granqvist, C.G.
Electrochromic materials: Microstructure, electronic bands, and optical properties
DOI: 10.1007/BF00331209
3) Results:
In your photo, the bleached state of your device is very dark. Please explain why your EC device has so low transparency? Can you improve it? What are the differences in the transparencies for colored and bleached states? Show it clearly in pictures (Fig2)
Impedance.
Your impedance spectra are not readable. You have shown the equivalent circuit, but there are no features visible that would stat for Rct, Rf, and CPE. Moreover, once you introduce constant phase elements, you have to explain what do you expect to be acting of capacitance and resistivity in your model. The data presented as-is in Fig4 must be revised. Please show the plots correctly (with Rct Rf features) and provide the resistance on the plots.
Please compare your materials with the state of art.
Author Response

(The authors gave the same response as above.)

Reviewer 3 Report
This work focused on the preparation and analysis of PADA gel electrolyte and then its possible electrochromic application is investigated. The work included sufficient characterization, with providing detailed experimental procedure. This work will be interesting for the reader of the journal. However, making the following changes are necessary before considering the manuscript for publication:
1- The novelty of this work should be clearly addressed in the introduction section, since many other similar works can be found in the literature.
2- There are very few keywords to describe the subject of the entire article, and more keywords should be added.
3- It is not easy to follow the introduction section since there is no link between different paragraphs and authors jumped from one subject to another randomly. Furthermore, the provide background information and addressing previous works in this field are very limited and not sufficient. In the last paragraph of the introduction section (page 2, line 49), authors addressed many achieved results, which is not an appropriate place to present your results and they all need to be removed. In steady of that, authors need to clearly highlight the importance and necessity of this study. Overall, the introduction section should re-constructed and re-written again in the light of given comments and suggestions.
4- There are many in appropriate sentence structural, grammatical mistakes and format errors throughout the manuscript. Such as spaces between values and units (page 2, line 74) “0.5ml of acrylic acid”, “1.25mg AIBN”, (page 3, line 110) “The difference between the two is the optical the modulation amplitude is 4.1%.”, … etc. Those mistakes should be avoided through carefully checking the entire manuscript.
5- Authors should discuss the reason for PADA-b gel electrolyte high transparency compared to PADA-a and PADA-c samples, rather than just describing Figure 3.
6- Use equations should be cited such as equation (1) and (2).
7- Why negative sign is put for the imaginary part of impedance (- ) in y-axis of Figure 4(a,b).
8- what is the importance of Figure 4 (b) since it does not reveal anything new and it can be removed.
9- In page 6, line 165 change “The electrical conductivity” to ionic conductivity.
10- authors should find the ionic conductivity of all sample meaning (PADA-a, and PADA-c), for better comparison between the prepared samples.
11- What was the thickness of the samples? How it was measured? How authors calculated the conductivity without addressing the thickness of the samples? This information should be added in details to the manuscript.
12- How authors drawn an equivalent circuit without fitting the experimental data? What were the values of Rs, CPE, Rf in the equivalent circuit?
Author Response

(The authors gave the same response as above.)

Round 2
Reviewer 1 Report
The manuscript has been improved according to my comments. It is suitable for publication now.
Author Response
Thank you very much!
Reviewer 2 Report
The authors certainly improved the Figures' quality and added relevant references. However, there are two major points are still must be addressed:
1) The EIS. Your Nyquist plot is still awkward. You have to repeat the measurements. The provided equivalent circuit makes no sense when the enlarged image is provided. Basically, you have no features for the elements that you mention. Please work with literature on EIS interpretation and either repeat the EIS measurement or provide relevant equivalent circuit
2) You have to explain how your stack can be implemented in real applications. If you rely on the reflectance changes, what would be your real working device and its application?
Author Response
We sincerely thank you all for spending precious time and efforts in examining this manuscript and greatly appreciate your insightful comments to make the paper better. The manuscript has been carefully revised, improved and verified to address the questions raised by the reviewers, and revisions have been marked with green for clarification. Please see the attachment.

Reviewer 3 Report
Dear Authors,
Thanks for considering the proposed suggestions and comments. I am satisfied with your corrections in the revised version of the manuscript.
Author Response
Thank you very much!
Round 3
Reviewer 2 Report
Authors answered all my questions and addressed relevant modifications in the text.